# Neural Gaffer: Relighting Any Object via Diffusion

**Haian Jin**[1]    **Yuan Li**[2]    **Fujun Luan**[3]    **Yuanbo Xiangli**[1]    **Sai Bi**[3]
**Kai Zhang**[3]    **Zexiang Xu**[3]    **Jin Sun**[4]    **Noah Snavely**[1]

[1]Cornell Tech, Cornell University  [2]Zhejiang University
[3]Adobe Research  [4]University of Georgia

Project Website: `https://neural-gaffer.github.io/`

## Abstract

Single-image relighting is a challenging task that involves reasoning about the complex interplay between geometry, materials, and lighting. Many prior methods either support only specific categories of images, such as portraits, or require special capture conditions, like using a flashlight. Alternatively, some methods explicitly decompose a scene into intrinsic components, such as normals and BRDFs, which can be inaccurate or under-expressive. In this work, we propose a novel end-to-end 2D relighting diffusion model, called *Neural Gaffer*[1], that takes a single image of any object and can synthesize an accurate, high-quality relit image under any novel environmental lighting condition, simply by conditioning an image generator on a target environment map, without an explicit scene decomposition. Our method builds on a pre-trained diffusion model, and fine-tunes it on a synthetic relighting dataset, revealing and harnessing the inherent understanding of lighting present in the diffusion model. We evaluate our model on both synthetic and in-the-wild Internet imagery and demonstrate its advantages in terms of generalization and accuracy. Moreover, by combining with other generative methods, our model enables many downstream 2D tasks, such as text-based relighting and object insertion. Our model can also operate as a strong relighting prior for 3D tasks, such as relighting a radiance field.

## 1    Introduction

Lighting plays a key role in our visual world, serving as one of the fundamental elements that shape our interpretation and interaction with 3D space. The interplay of light and shadows can highlight textures, reveal contours, and enhance the perception of shape and form. In many cases, there is a desire to relight an image—that is, to modify it as if it were captured under different lighting conditions. Such a relighting capability enables photo enhancement (e.g., relighting portraits [18; 67]), facilitates consistent lighting across various scenes for filmmaking [59; 22], and supports the seamless integration of virtual objects into real-world environments [38; 19].

However, relighting single images is a challenging task because it involves the complex interplay between geometry, materials, and illumination. Hence, many classical model-based inverse rendering approaches aim to explicitly recover shape, material properties, and lighting from input images [2; 75; 47; 83; 87], so that they can then modify these components and rerender the image. These methods often require multi-view inputs and can suffer from: 1) *model limitations* that prevent faithful estimation of complex light, materials, and shape in real-world scenes, and 2) *ambiguities* in determining these factors without strong data-driven priors.

To circumvent these issues, image-based relighting techniques [18; 72; 76] avoid explicit scene reconstruction and focus on high-quality image synthesis. However, some image-based methods

---

[1]In film and television crews, the gaffer is responsible for managing lighting, including associated resources such as labor, lighting instruments, and electrical equipment

38th Conference on Neural Information Processing Systems (NeurIPS 2024).

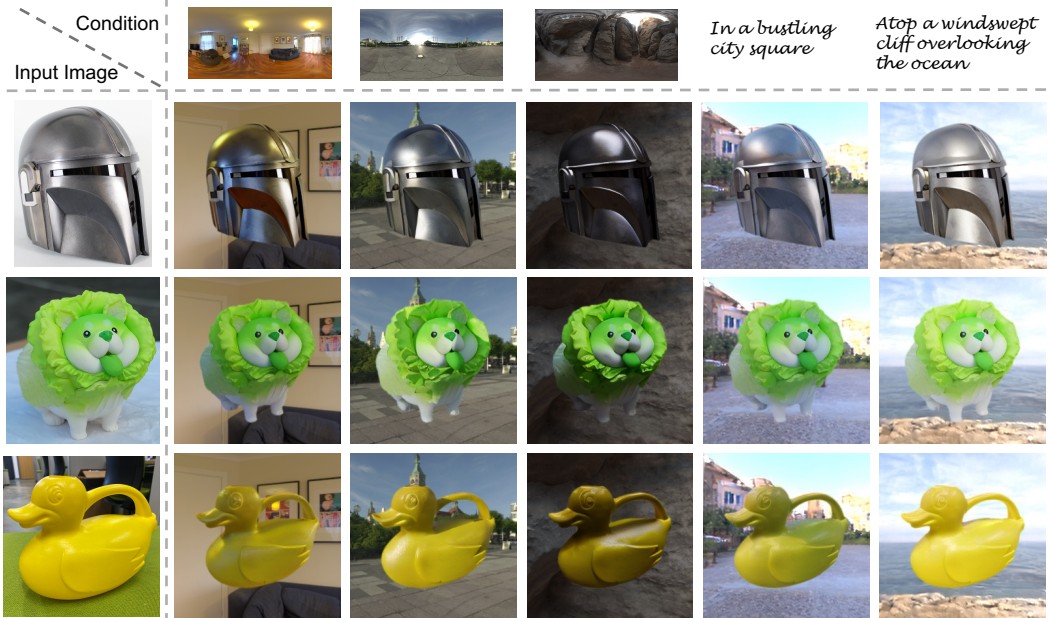

Figure 1: **Single-image relighting results on real data.** Neural Gaffer supports single-image relighting for various input images under diverse lighting conditions, using either image-conditioned input (i.e., an environment map) or text-conditioned input (i.e., a description of the target lighting). These results demonstrate our model's capability to adapt to diverse lighting scenarios while preserving the visual fidelity of the original objects. Our relighting results remain consistent with the lighting rotating. Please see the supplementary webpage for additional video results produced from real input images.

[18; 58; 76; 44; 3] require inputs under multiple lighting conditions or even sophisticated capture setups (e.g., a light stage [18]), while recent deep learning-based techniques [67; 51; 89] enable single-view relighting but often work only on specific categories like portraits.

In this paper, we propose a novel category-agnostic approach for single-view relighting that tackles the challenges mentioned above. We introduce an end-to-end 2D relighting diffusion model, named *Neural Gaffer*, that takes a single image of any object as input and synthesizes an accurate and high-quality relit image under any environmental lighting condition, specified as a high-dynamic range (HDR) environment map. In contrast to previous learning-based single-view models that are trained for specific categories, we leverage powerful diffusion models, trained on diverse data, and enable relighting of objects from arbitrary categories. Unlike prior model-based approaches, our diffusion-based data-driven framework enables the model to learn physical priors from an object-centric synthetic dataset featuring physically-based materials and High Dynamic Range (HDR) environment maps. This facilitates more accurate lighting effects compared to traditional methods.

Our model exhibits superior generalization and accuracy on both synthetic and real-world images (See Fig. 1). It also seamlessly integrates with other generative methods for various 2D image editing tasks, such as object insertion. Neural Gaffer also can serve as a robust relighting prior for neural radiance fields, facilitating a novel two-stage 3D relighting pipeline. Hence, our diffusion-based solution can relight any object under any environmental lighting condition for both 2D and 3D tasks. By harnessing the power of diffusion models, our approach generalizes effectively across diverse scenes and lighting conditions, overcoming the limitations of prior model-based and image-based methods.

## 2   Related Work

**Diffusion models.** Recently, diffusion-based image generation models [26; 49] have dominated visual content generation tasks. Stable Diffusion [60] not only generates impressive results in the text-to-image setting, but also enables a wide range of image applications. ControlNet [84] adds trainable neural network modules to the U-Net structure of diffusion models to generate images

conditioned on edges, depth, human pose, etc. Diffusion models can also change other aspects of an input image [45; 9], such as image style, via image-to-image models [52]. ControlCom [80] and AnyDoor [14] are image composition models that can blend a foreground object into a target image. Huang et al. provide a comprehensive review of diffusion-based image editing [28].

Recent work has also shown that, despite being trained on 2D images, other kinds of information can be unlocked from diffusion models. A notable example is Zero-1-to-3 [42], which uncovers 3D reasoning capabilities in Stable Diffusion by fine-tuning it on a view synthesis task using synthetic data, and conditioning the model on viewpoint. Similar to the spirit of Zero-1-to-3 and its follow-up work (such as [63; 27]), our work unlocks the latent capability of diffusion models for relighting.

**Single-image relighting.** Relighting imagery is a difficult task that requires an (explicit or implicit) understanding of geometry, materials, lighting, and their interaction through light transport. Single-image relighting is especially challenging due to its ill-posedness; there are infinite explanations of a single image in terms of underlying factors. Classic single-image methods, like SIRFS [2], focus on explicit estimation of scene properties, via optimization with hand-crafted priors. More recent learning-based methods can learn such priors from data, but either focus on specific image types, such as outdoor scenes [73; 69; 78; 40; 21], portraits [53; 71; 64; 67; 48; 51; 23; 56], or human bodies [31; 37; 29], or require special capture conditions like camera flash [39; 61]. Single-image relighting methods for general objects, with realistic appearance, materials, and lighting, remain elusive. In addition, most recent single-image relighting methods [51; 39; 61; 12] still explicitly decompose an image into components, then reconstruct it with modified lighting. This strategy allows the components to be explicitly controlled, but also limits their realism based on the representational ability of the chosen components, such as the Disney BSDF [11] or Phong reflection model [10]. In contrast, similar to [67; 79], we achieve object relighting without an explicit scene decomposition. By training an image-based and lighting-conditioned diffusion model, we can relight any given object from a single photo.

**3D relighting and inverse rendering.** Inverse rendering methods involve estimating 3D models that can be relit under different lighting conditions, typically from multiple images. Recent advances in 3D, such as NeRF [46] and 3D Gaussian Splatting [32], have shown remarkable novel view synthesis results. However, these representations are not themselves relightable as the physical light transport is baked into the radiance field. Efforts have been made toward creating relightable 3D representations, such as PhySG [83], NeRFactor [87], TensoIR [30] and others [6; 8; 7; 5; 4; 82; 88; 35; 66; 77; 81]. These methods often rely on pure optimization to minimize a rendering loss, sometimes incorporating hand-crafted material and/or lighting regularization terms (e.g., smoothness or sparsity) to prevent overfitting. However, these approaches often struggle with specular highlights and glossy materials. We show that our proposed Neural Gaffer method can be applied in an inverse rendering framework, serving as a powerful prior and regularizer. This is because our model has been trained on a large dataset encompassing diverse lighting and material conditions, effectively learning these priors from high-quality data. Our model improves 3D relighting quality, while baselines tend to produce artifacts due to the baking of lighting information into the color field.

## 3 Method

Neural Gaffer is a generalizable image-based 2D diffusion model for relighting general objects; we finetune a pretrained image-conditioned diffusion model [42] with a synthetic relighting dataset to fulfill this goal. In Sec. 3.1, we describe how we construct a large-scale high-quality relighting dataset from Objaverse [20]. Then, in Sec. 3.2, we discuss the design and training of a 2D diffusion model to support zero-shot single-image relighting. Finally, we showcase applications of our model in downstream tasks, including 2D object insertion and 3D radiance field relighting tasks.

### 3.1 RelitObjaverse Dataset

Capturing ground-truth lighting for real object images is both time-consuming and challenging. Existing real-world relighting datasets [41; 36; 70] are constrained to a very limited number of objects. To address this limitation, we opt to render a large-scale synthetic object relighting dataset.

We use Objaverse [20] as our data source, which comprises about 800K synthetic 3D object models of varying quality. To avoid 3D object models with poor geometry or texture, we developed a series of

filtering mechanisms. Specifically, we first utilize a classifier trained on a manually annotated subset to categorize and filter out the 3D objects with low geometry and texture quality. In addition, we also design a set of filtering rules based on 3D object attributes, such as geometry bounding boxes, BRDF attributes, and file size, eventually yielding a subset (of ∼90K objects) of the original Objaverse with high quality. To ensure abundant lighting variations, we collected 1,870 HDR environment maps from the Internet, and augmented them with horizontal flipping and rotation, resulting in ∼30K different environment maps in total.

We render images with the Cycles renderer from Blender [17]. During rendering, we randomly sample 12 camera poses for each object. For each viewpoint, we render 16 images under different environment maps. In contrast to existing datasets that only render with environment maps [16; 62], we additionally render an image under a randomly set area light source to increase lighting diversity. In total, for each object, there are $12 \times (16 + 1) = 204$ images rendered under different lighting conditions and from different viewpoints. Our final synthetic relighting dataset has about 18.4M $512 \times 512$ rendered images with ground-truth lighting. We will release our full relighting dataset, named the RelitObjaverse dataset, upon acceptance to facilitate reproducibility.

## 3.2 Lighting-conditioned 2D Relighting Diffusion Model

Given a single object image $x \in \mathbb{R}^{H \times W \times 3}$ and a new lighting condition $E \in \mathbb{R}^{H' \times W' \times 3}$ (represented as an HDR environment map), our goal is to learn a model $f$ that can synthesize a new image $\hat{x}_E$ with correct appearance changes under the new lighting, while preserving the object's identity:

$$\hat{x}_E = f(x, E). \tag{1}$$

We fine-tune an image-to-image diffusion model using our rendered data. In particular, we adopt the pre-trained Zero-1-to-3 model [42] and discard its components and weights relevant to novel-view generation, repurposing the model for relighting. To incorporate lighting conditions, we extend the input latents of the diffusion model, concatenating the encoded input image and the environment map with the denoising latent. We introduce two design decisions that are critical for the effective integration of lighting into the latent diffusion model.

**Environment map rotation.** In general, each pixel of an environment map $E$ corresponds to a 3D direction in the absolute world coordinate frame, whereas our relighting is performed in image space, that is, locked to the camera's coordinate frame. Therefore, we need a way for a user to specify the relationship between the environment map and camera coordinate frames. One option is to concatenate the relative 3D rotation to the embedding, similar to how Zero-1-to-3 [42] encodes the relative camera pose for the novel view synthesis task. However, we observe that such a design makes it harder for the network to learn the relighting task, perhaps because it needs to interpret the complex relationship between 3D rotation parameters and the desired rotated lighting. Our solution is instead to rotate the target environment map to align with the target camera coordinate frame, prior to feeding it to the model. After rotation, each pixel of the rotated environment map corresponds to a fixed lighting direction in the target camera coordinate frame.

**HDR-LDR conditioning.** Pixel values in the HDR environment map have a theoretical range of $[0, +\infty)$, and large pixel values can lead to unstable diffusion training. To rescale it to $[0, 1.0]$, a naive idea is to divide it by its maximum value, but this leads to a loss of total energy information. Also, if the map has a very bright region (e.g., the sun), then most of the other values will approach 0 and the environment map will be too dark after division, causing a loss of lighting detail. To solve this, we split the HDR environment map $E$ into two maps: $E_L$ is an LDR representation of $E$, computed via standard tone-mapping [57; 24], and $E_H$ is a nonlinearly normalized representation of $E$, computed by a logarithmic mapping and divided by the new maximum value. $E_L$ encodes lighting detail in low-intensity regions, which is useful for generating relighting details(such as reflection details). $E_H$ maintains the lighting information across the initial spectrum. Inputting $E_L$

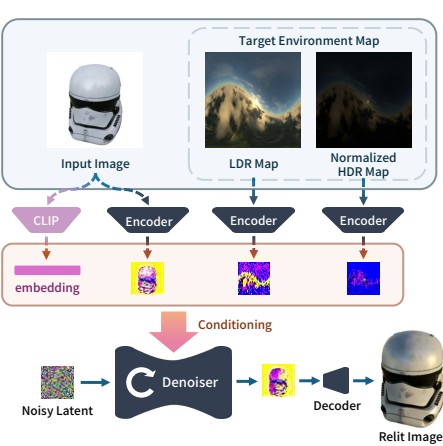

Figure 2: **Model architecture.** Neural Gaffer is an img2img latent diffusion model conditioned on the input image and rotated lighting maps.

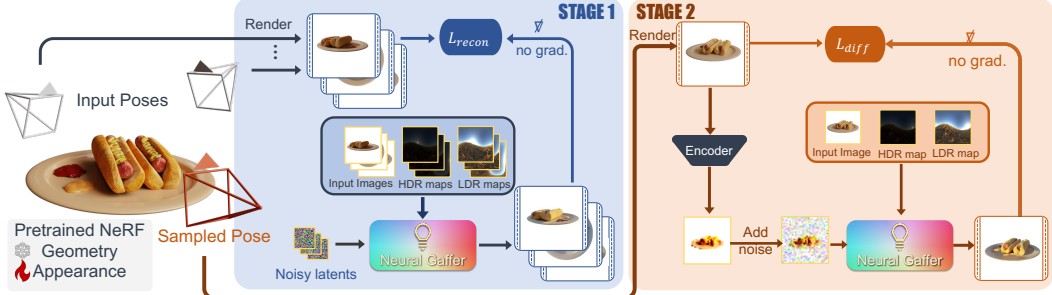

Figure 3: **Relighting a 3D neural radiance field.** Given an input NeRF and a target environmental lighting, in **Stage 1**, we use Neural Gaffer to predict relit images at each predefined camera viewpoint. We then tune the appearance field to overfit the multi-view relighting predictions with a reconstruction loss. In **Stage 2**, we further refine the appearance of the coarsely relit radiance field via the diffusion guidance loss. Using this pipeline, we can relight a NeRF model in minutes.

and $E_H$ together to the diffusion model means that the network can reason effectively about the full original energy spectrum. Additionally, our design helps to achieve a balanced exposure in the generated output, preventing areas from appearing too dark or too washed out after normalization due to extreme brightness.

**Diffusion model architecture and training.** Our latent diffusion model comprises an encoder $\mathcal{E}$, a denoiser U-Net $\epsilon_\theta$, and a decoder $\mathcal{D}$. During each training iteration, given the paired input-target images $x$ and $\hat{x}_E$, and the target lighting condition $E$, we (1) rotate $E$ into the target camera coordinate frame based on the specified lighting direction, forming a rotated map $\hat{E}$; (2) transform $\hat{E}$ into an LDR environment map $\hat{E}_L$ and a normalized HDR environment map $\hat{E}_H$; (3) and finally, encode $\hat{E}_L$, $\hat{E}_H$ and the input image $x$ with the encoder $\mathcal{E}$. The concatenation of the encoded latent maps are used to condition the denoising U-Net $\epsilon_\theta$. Fig. 2 depicts our 2D relighting system. We fine-tune the latent diffusion model with the following objective:

$$\min_\theta \ \mathbb{E}_{z \sim \mathcal{E}(x), t, \epsilon \sim \mathcal{N}(0,1)} ||\epsilon - \epsilon_\theta(z_t, t, \hat{\mathcal{E}}, c(x))||_2^2, \tag{2}$$

where $t \sim [1, 1000]$ is the diffusion time step, $\hat{\mathcal{E}} = \{\mathcal{E}(x), \mathcal{E}(\hat{E}_L), \mathcal{E}(\hat{E}_H)\}$ represents the union of encoded latents for $x$, $\hat{E}_L$ and $\hat{E}_H$, and $c(x)$ represents the CLIP embedding of the input image $x$. Example 2D relighting results from our trained model, applied to real images, are shown in Fig. 1.

### 3.3 Relighting a 3D radiance field with diffusion prior guidance

We can utilize Neural Gaffer as a prior for 3D relighting. In particular, given a 3D neural radiance field consisting of a density field $\mathcal{G}_\sigma$ and a view-dependent appearance field $\mathcal{G}_a$ [46], we consider the task of relighting it using our 2D relighting diffusion model as a prior. We find that simply relighting the input NeRF with the popular Score Distillation Sampling (SDS) method [55] leads to over-saturated and unnatural results, as shown in Fig. 8. We instead propose a two-stage pipeline (Fig. 3) to achieve high-quality 3D relighting in minutes. In the first stage, we use our 2D relighting diffusion model to relight input views and overfit the original NeRF's appearance field, giving us coarsely relit renderings at sampled viewpoints. In the second stage, our model is used to further refine the appearance details with a diffusion loss and an exponential annealed multi-step scheduling.

**Stage 1: Coarse relighting with reconstruction loss.** Given a NeRF with parameters $\psi$, the multi-view input images $x_{\text{input}} = \{x_i\}_{i=1}^N$ and the target lighting $E_{\text{target}}$, we first use our diffusion model $f$ to generate multi-view relighting prediction images $\hat{x}_{\text{relit}} = \{\hat{x}_i\}_{i=1}^N$, where $\hat{x}_i = f(x_i, E_{\text{target}})$.

Then, we freeze the density field $\mathcal{G}_\sigma$ of the NeRF, and only optimize the appearance field of the input NeRF with reconstruction loss $L_{\text{recon.}}$, aiming to minimize the reconstruction error between a rendered image $x = x(\psi, \pi_{\text{input}})$ and the relit image $x_{\text{relit}}$ at pose $\pi_{\text{input}}$:

$$\mathcal{L}_{\text{recon.}}(\psi) = ||(x(\psi, \pi_{\text{input}}) - x_{\text{relit}}||_2^2. \tag{3}$$

After this stage, we get an initial blurry and noisy relighting result.

**Stage 2: Detail refinement with diffusion guidance loss.** The results from the first stage can be blurry and can exhibit noise because the multi-view relighting predictions results may be inconsistent.

We further refine the relit NeRF's appearance in Stage 2. We found that directly using the SDS loss [55] for refinement leads to noisy and over-saturated results (see Fig. 8). Inspired by [68; 74; 45], we treat the blurry rendered images from Stage 1 as an intermediate denoising result. We first render an image $\hat{x}$ under the sampled camera pose $\pi_{\text{sampled}}$. We encode and perturb it to a noisy latent $\hat{z}_t(\hat{x}, t^\star)$ with an intermediate noise level $t^\star$, and initiate the multi-step DDIM [65] denoising process for $\hat{z}_t(\hat{x}, t^\star)$ from $t^\star$, yielding a latent sample $\hat{z}_0$. This latent is decoded to an image $\hat{x}_{\text{relit}}(t^\star) = D(\hat{z}_0)$, which we use to supervise the rendering with the following diffusion guidance loss:

$$\mathcal{L}_{\text{diff}}(\psi) = w_1 \|\hat{x} - \hat{x}_{\text{relit}}(t^\star)\|_1 + w_2 L_p(\hat{x}, \hat{x}_{\text{relit}}(t^\star)), \tag{4}$$

where $L_p$ is the perceptual distance LPIPS [86] and $w_1$ and $w_2$ are scale factors to balance the two terms. We exponentially decrease $t^\star$ during optimization.

## 4 Experiments

### 4.1 Implementation details

**Diffusion model training.** We fine-tune our model starting from Zero123's [42] checkpoint and discard its original linear projection layer for image embedding and pose. We only fine-tune the UNet of the diffusion model and freeze other parts. During fine-tuning, we use a reduced image size of $256 \times 256$ and a total batch size of 1024. Both the LDR and normalized HDR environment maps are resized to $256 \times 256$. We use AdamW [43] and set the learning rate to $10^{-4}$ for training. We fine-tune our model for 80K iterations on 8 A6000 GPUs for 5 days.

**3D relighting.** Given a neural radiance field reconstructed from multi-view inputs (we use TensoRF [13]), we first freeze its density field and then optimize the input NeRF's appearance field for 2,500 iterations in Stage 1, with a ray batch size of 4096. In Stage 2, we set the initial timestep $t^\star$ to decrease from 0.4 to 0.05 exponentially. $w_1$ and $w_2$ in Eqn. 4 are set to be 0.5 for all of our test cases. We optimize the results from Stage 1 for 500 iterations, and for each iteration, we sample one view to compute the diffusion guidance loss in Eqn. 4. Stage 1 takes about 5 minutes and Stage 2 takes about 3 minutes on a single A6000 GPU.

**Downstream 2D tasks.** Our model supports downstream 2D image editing tasks, such as text-based relighting and object insertion. For text-based relighting, given a text prompt, we first use Text2Light [15] to generate an HDR panorama, and, using that as an environment map, relight images using our diffusion model. Fig. 1 (right two columns) shows example results; more are on our project webpage. For object insertion, given an object image and a target (perspective) scene image, we segment the foreground object with SAM [34], estimate lighting from the target background image with DiffusionLight [54], perform relighting, and finally generate a shadow attached to the object using [50]. Our workflow can preserve the object's identity and achieve high-quality visual quality.

### 4.2 Baselines and metrics

We use common image metrics including PNSR, SSIM, and LPIPS [86] to evaluate relighting results quantitatively. In addition, since there exists an ambiguity between lighting intensity and object color, we also compute channel-aligned results as in [87; 88], where each channel of the relighting prediction is rescaled to align the average intensity with the ground-truth. We compute metrics for both the raw and the channel-aligned results for quantitative evaluation.

For the single-image relighting task, there are very few prior works that can handle general objects under natural illumination. We compare to a very recent work DiLightNet [79], which trains a lighting-conditioned ControlNet [84] to relight images, which is achieved by conditioning its ControlNet with a set of rendered images using predefined materials, estimated geometry of the input image, and target lighting. Per our request, the authors ran their model on our validation data. We select 48 high-quality objects from Objaverse as validation objects, which are unseen during training. We render each object under 4 different camera poses. For each camera, we randomly sample 12 unseen environment maps to render the target relit images, and one additional environment map to render the input. We also compare with IC-Light [85]. Since our lighting conditioning is different from its setting (they use background images), we only provide qualitative comparisons in our appendix and project webpage. For relighting 3D radiance fields, we compare with two 3D inverse rendering methods: TensoIR [30] and NVDIFFREC-MC [25]. We select 6 objects from the NeRF-Synthetic

dataset [46] and Objaverse [20] for testing. For each object, 100 camera poses are sampled to generate training images and 20 camera poses are sampled for testing images. Each pose is rendered under four distinct, previously unseen environmental lighting conditions. One lighting condition is designated for training images, while the remaining three are reserved for relighting testing.

### 4.3 Results analysis

**Comparing single-image relighting with DiLightNet [79].** As shown in Fig. 4, we evaluate across a diverse set of unseen objects and unseen target lighting conditions. Our approach showcases superior fidelity to the intended lighting settings, maintaining consistent color accuracy and preserving intricate details. Compared with DiLightNet [79], our model can generate more accurate highlights, shadows, and high-frequent reflections. These results demonstrate our model's ability to accurately understand the underlying physical properties of the object in the input image, such as geometry and materials, and faithfully reproduce the desired relighting appearance. Quantitative results are in Tab. 1.

**Single-image relighting on real data.** We evaluate our single-image relighting model on a set of real-world input photos (Fig. 1), under either image-conditioned (environment map) or text-conditioned (text description of the desired target lighting) target illumination. As shown in the figure, our model effectively adapts to diverse lighting conditions and produces realistic, high-fidelity relighting results under novel illumination. In addition, our relighting results are consistent and stable with the lighting condition rotating. Please refer to our webpage to get more video results of more real data.

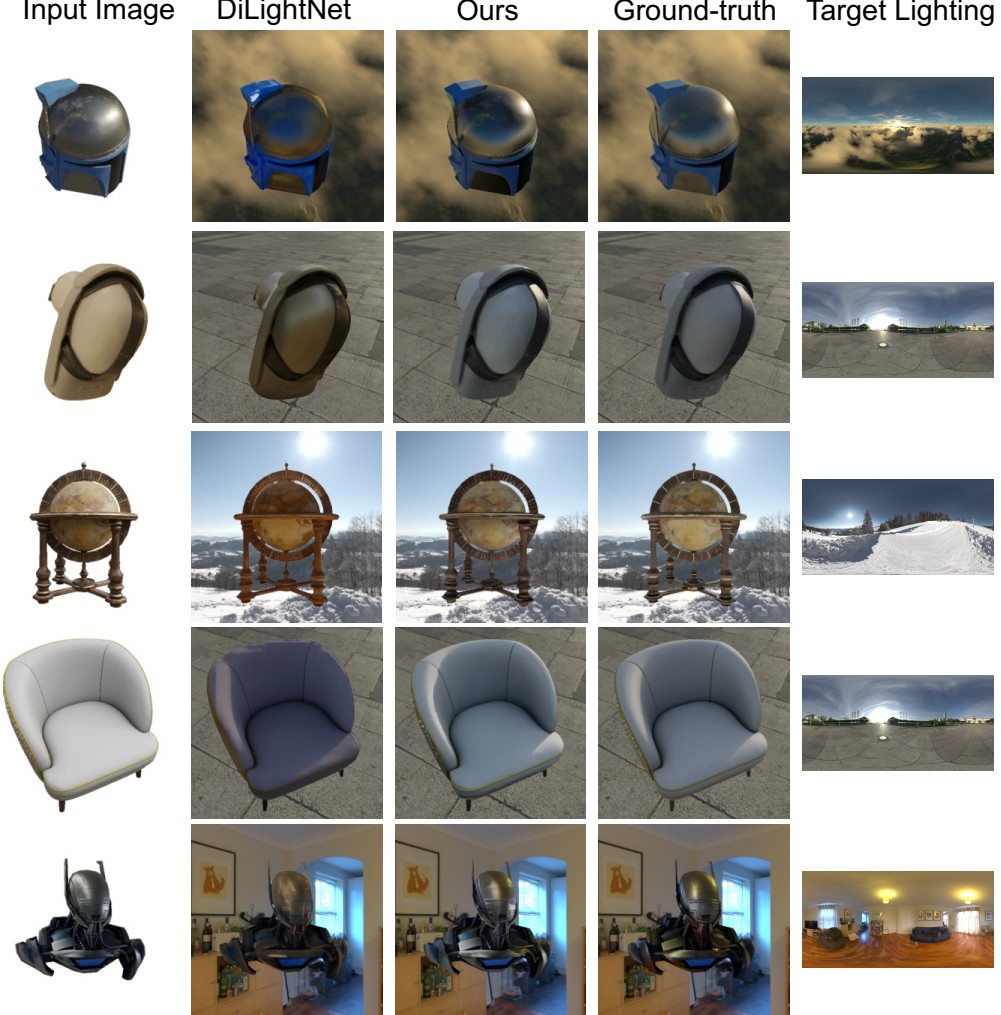

Figure 4: **Single-image relighting comparison with DiLightNet [79] under diverse lighting.** Our method demonstrates superior fidelity to the target lighting, maintains more consistent color and detail, and can generate more accurate highlights, shadows, and high-frequent reflections.

Table 1: Single-image relighting quantitative comparison.

| Method | Relighting | | | Channel-aligned Relighting | | |
|---|---|---|---|---|---|---|
| | PSNR ↑ | SSIM ↑ | LPIPS ↓ | PSNR ↑ | SSIM ↑ | LPIPS ↓ |
| Ours | **26.706** | **0.927** | **0.041** | **29.829** | **0.939** | **0.028** |
| DiLightNet [79] | 24.823 | 0.918 | 0.056 | 26.931 | 0.926 | 0.050 |

Table 2: 3D relighting quantitative comparison.

| Method | PSNR ↑ | SSIM ↑ | LPIPS ↓ |
|---|---|---|---|
| Ours | **29.11** | **0.930** | **0.039** |
| TensoIR [30] | 25.80 | 0.878 | 0.106 |
| NVDIFFREC-MC [25] | 28.25 | 0.915 | 0.082 |

**Object insertion.** Our diffusion model exhibits versatility when combined with other generative models for downstream 2D tasks, like object insertion. In Fig. 5, we take a foreground image containing the object of interest and insert the object into a background image depicting the desired target lighting condition. Our method surpasses baseline approaches by better preserving the identity of the inserted objects and synthesizing more natural illumination, seamlessly integrating objects into the background scene.

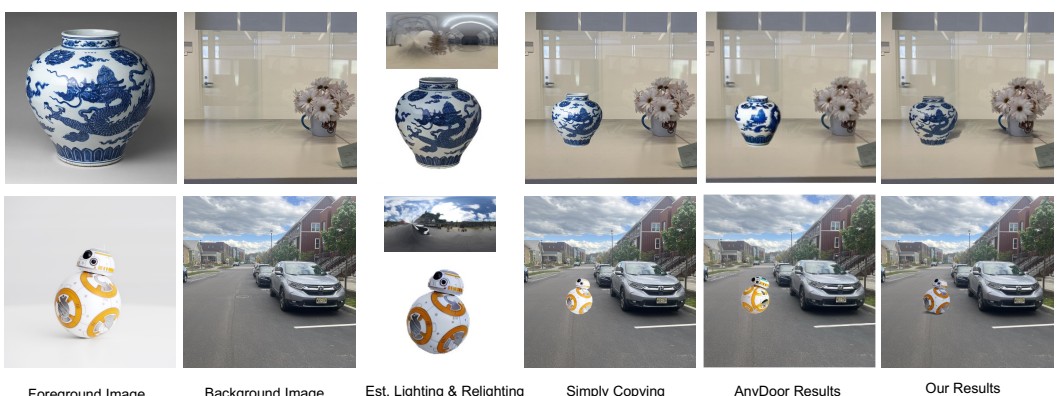

Foreground Image    Background Image    Est. Lighting & Relighting    Simply Copying    AnyDoor Results    Our Results

Figure 5: **Object insertion.** Our diffusion model can be applied to object insertion. Compared with Any-Door [14], our method better preserves the identity of the inserted object and achieves higher-quality results.

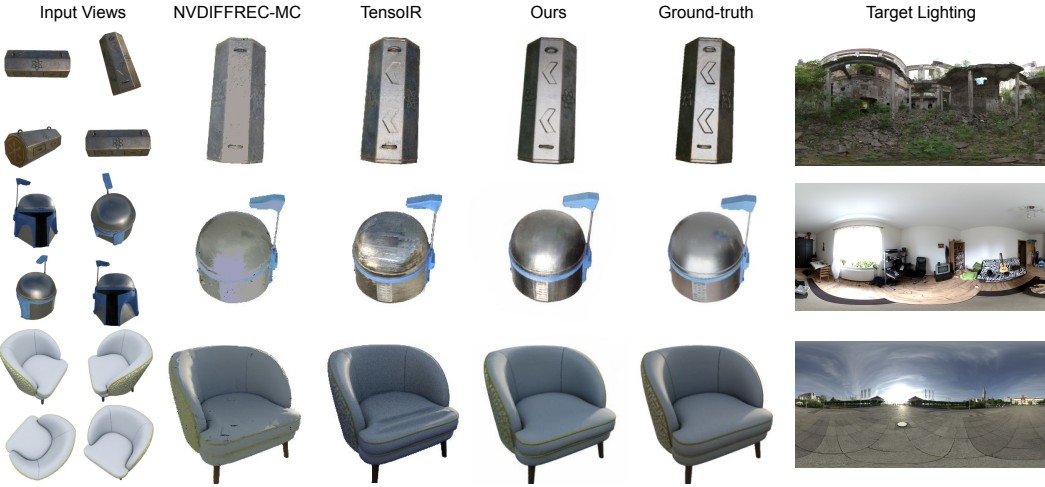

Input Views    NVDIFFREC-MC    TensoIR    Ours    Ground-truth    Target Lighting

Figure 6: **Relighting 3D objects.** We take multi-view images and a target lighting condition as input and compare our method's relighting results to NVDIFFREC-MC [25] and TensoIR [30]. (The shown images are channel-aligned results.) Lacking strong priors, the baselines tend to miss specular highlights (NVDIFFREC-MC), bake the original highlights (TensoIR), or produce artifacts (NVDIFFREC-MC and TensoIR). In contrast, our method implicitly learns to model the complex interplay of light and materials from the training process of our relighting diffusion model, resulting in more accurate relighting results with fewer artifacts. Video comparisons are available on our project webpage.

**Relighting 3D objects.** Fig. 6 shows the results of our two-stage method for relighting 3D objects. Given an input radiance field(NeRF) representing a 3D object, we first tune NeRF for coarse relighting in Stage 1, followed by a finer relighting result in Stage 2 that improves fidelity. We also compared to conventional inverse rendering methods such as NVDIFFREC-MC [25] and TensoIR [30], reporting quantitative results in Tab. 2 and showing qualitative comparisons in Fig. 6 and Fig. 13.

## 4.4 Ablation Study

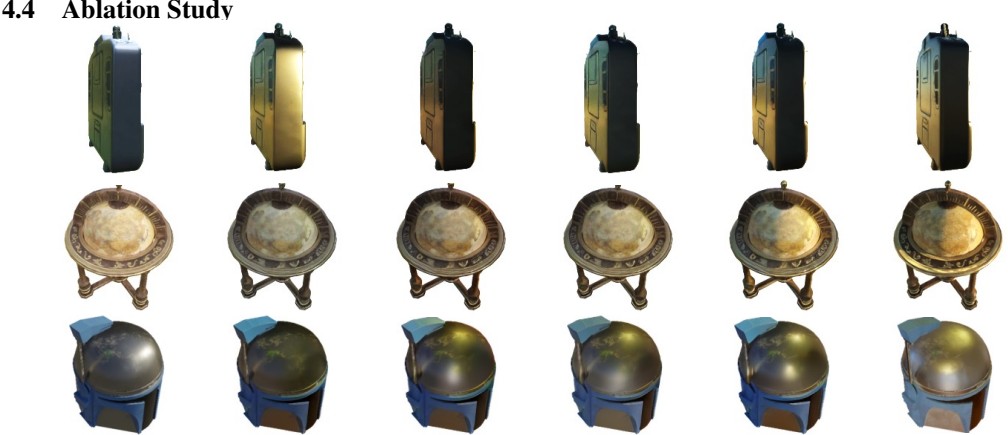

|  |  |  |  |  |  |
|---|---|---|---|---|---|
| Input image | w/o Rotation | w/o LDR map | w/o HDR map | Ours | Ground-truth |

Figure 7: **Ablations on the lighting conditioning designs** for our single-image relighting diffusion model.

**Ablation studies for alternate lighting conditioning designs for our diffusion model.**

As ablations, we evaluated different lighting condition-ing designs for our diffusion model: 1) only inputting one environment map rather than both the rotated LDR and normalized HDR map, and 2) controlling the light-ing direction by adding the relative 3D rotation between the default environment map coordinate frame and the target camera frame to the U-Net CLIP embedding branch without rotating the environment maps explicitly.

Table 3: Ablations on diffusion model designs.

| Method | PSNR ↑ | SSIM ↑ | LPIPS ↓ |
|---|---|---|---|
| Ours | **26.052** | **0.923** | **0.035** |
| Ours w/o LDR map | 25.503 | 0.920 | 0.045 |
| Ours w/o HDR map | 25.822 | 0.922 | 0.036 |
| Ours w/o rotation | 24.455 | 0.915 | 0.051 |

Due to limited computational resources, when performing ablations for the model design, we train the different models with 2 A6000 GPUs for 48K iterations. We provide a quantitative comparison between them in Tab. 3 and a qualitative comparison in Fig. 7, showcasing the effectiveness of our design. As shown in Fig 7, controlling the lighting direction by only adding the relative 3D lighting rotation to the U-Net CLIP embedding branch can't get the relighting results with correct lighting directions for all testing cases. Without inputting the LDR map, the model fails to generate the correct reflection for the first test case. Without the HDR map, it cannot produce the correct specular highlights for the third test case. Only when both the LDR and HDR maps are provided can the model accurately reason about the full original energy spectrum and generate correct relighting results for the second test case. Our full model achieves the best results across different ablation experiments.

**Ablation studies for relighting 3D radiance fields.** As shown in Fig. 8, if we directly relight the input neural radiance field using an SDS loss [55] without our two-stage pipeline, we cannot accurately reproduce highlights and reflections in the first scene, generate precise shadows in the

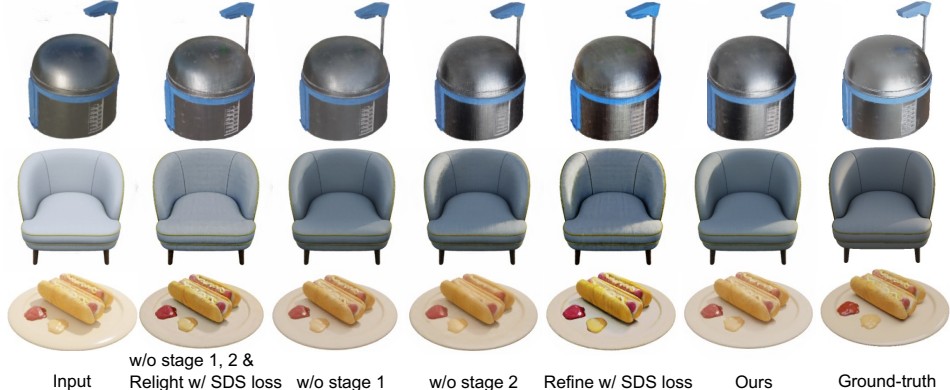

|  |  |  |  |  |  |  |
|---|---|---|---|---|---|---|
| Input | w/o stage 1, 2 & Relight w/ SDS loss | w/o stage 1 | w/o stage 2 | Refine w/ SDS loss | Ours | Ground-truth |

Figure 8: **Ablations on methods for relighting 3D radiance fields.** Our full model achieves the most accurate and visually realistic relighting results among all variants. Please zoom in to see detailed differences across ablation variants.

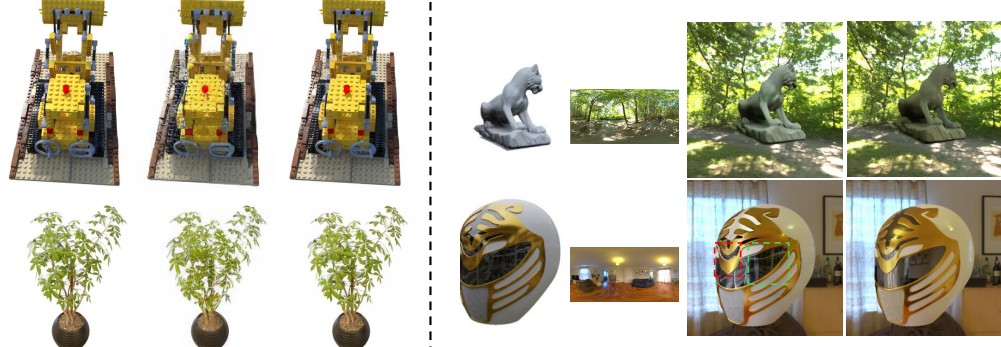

| Input Image | VAE (res-256) | VAE (res-512) | Input Image | Target EnvMap | Pred Image | Target Image |

Figure 9: **Limitation Analysis.** The left side shows the encoding-decoding results using the VAE [33] in our base diffusion model [1; 42]. These results indicate that the VAE struggles to preserve identity for objects with fine details at low resolution ($256 \times 256$), which in turn affects the performance of our model on such objects. On the right side, we present several failed cases of our method in Objaverse instances.

second sofa scene, or remove shadows in the third hotdog scene. Stage 1 ensures overall relighting accuracy, while Stage 2 helps reduce noise and leads to more detailed results. Without Stage 1, we cannot generate the correct specular highlight for the first scene, and the details of the shadows on the sofa scene's backrest become noisy and less accurate. Without Stage 2, the appearance of the relit scenes becomes noisier and blurrier, especially in the sofa scene. Replacing our diffusion guidance loss (Eqn. 4) in Stage 2 with an SDS loss results in unnatural, noisy, and over-saturated relighting. Our full model consistently produces the most accurate and visually realistic results across all test variants. Please zoom in to see detailed differences across ablation variants.

## 5   Limitations

While our model can achieve overall consistent relighting results under rotating or changing lighting conditions, our results can still exhibit minor inconsistencies since the model is generative.

Given the high resource demands of data preprocessing (specifically, rotating the HDR environment map) and model training, and considering our limited university resources, we trained the model at a lower image resolution of $256 \times 256$. In Fig. 9, we use VAE [33] in our base diffusion model [1; 42] to encode input images into latent maps, and then directly decode them. We found that VAE struggles to preserve identity for objects with fine details even from latent maps encoded from the input images at this resolution, which in turn results in many relighting failure cases at this resolution. Finetuning our model at a higher resolution will greatly help solve this issue.

We observe two key failure cases when relighting Objaverse Instances, as shown on the right side of Fig. 9. First, our model struggles with color ambiguity in diffuse objects, leading to inaccuracies in relit colors (first row). This issue arises from inherent ambiguities in the input image, where color could be attributed to either material or lighting. Including the background of the input image may help the model better infer the lighting conditions of input images and address this challenge. We give more discussions in the Appendix. C. Second, while the model generates low-frequency highlights well, it often misses high-frequency reflections in relit images (second row).

In addition, we discuss our limitations regarding portrait relighting in the Appendix. B.

## 6   Conclusion

We proposed an end-to-end 2D relighting diffusion model, Neural Gaffer, that addresses the long-standing challenge of single-image relighting through a data-driven framework. Our model synthesizes accurate and high-quality relighted images from a single input image under any environmental lighting condition, demonstrating superior generalization and accuracy on both synthetic and real-world datasets. Neural Gaffer not only enhances various 2D tasks, such as text-based relighting and object insertion but also serves as a strong relighting prior for 3D tasks, enabling a simplified two-stage 3D relighting pipeline. By overcoming the limitations and ambiguities of traditional model-based inverse rendering approaches, our diffusion-based solution extends the applicability of relighting techniques to a broad range of practical applications.

**Acknowledgments.** We thank Minghua Liu, Chao Xu, et al from Prof. Hao Su's group at UCSD for assisting with the Objaverse filtering process. We thank Xinrui Liu at Cornell for helping with data capture and shadow generation for the object insertion application. We thank Chong Zeng at Zhejiang University for helping run his model (DiLightNet) on our validation data

This work was done while Haian Jin was a full-time student at Cornell. The selection of data and the generation of all figures and results was led by Cornell University. This work was funded in part by the National Science Foundation (IIS-2211259). Jin Sun is partly supported by a gift from Google.

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

# A    Comparison with IC-Light

In this section, we conduct a comparison with a recent single-image relighting method, IC-Light [85]. In Fig. 10, we present a comparative analysis of relighting results, focusing on the consistency of specular highlights in comparison to IC-Light. Our method consistently adjusts the specular highlights in accordance with the rotation of the lighting, ensuring a coherent relighting effect. In contrast, IC-Light demonstrates inconsistent movement of highlights, with certain regions showing minimal changes. For additional video comparisons, please visit our supplemental webpage.

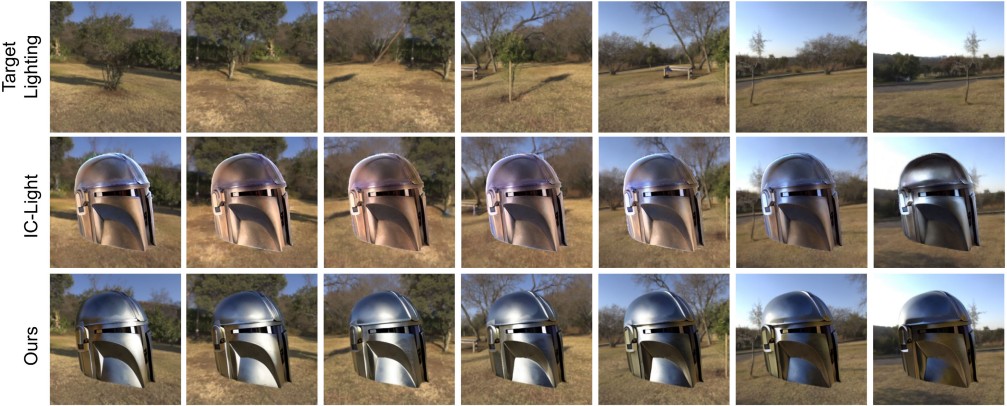

Figure 10: **Comparison of single-image relighting with IC-Light [85].** In this experiment, we compare relighting results and evaluate the consistency of specular highlights against IC-Light [85]. Each column represents a uniformly rotated version of the same environment map image, progressing from left to right. Our method consistently adjusts highlights, faithfully reproducing the object under novel lighting conditions. In contrast, IC-Light [85] exhibits inconsistent highlight movement, with certain regions changing just slightly. Further comparisons are available in videos provided on the supplemental webpage.

# B    Portrait Relighting Results

Our model is fine-tuned on object data, therefore, while it is category-agnostic, it might not achieve the highest quality results on specific domains—for instance, for the case of portrait relighting, methods that are specifically trained for portrait photos may produce higher quality results. We show our results on the portrait in Fig. 11 of the appendix.

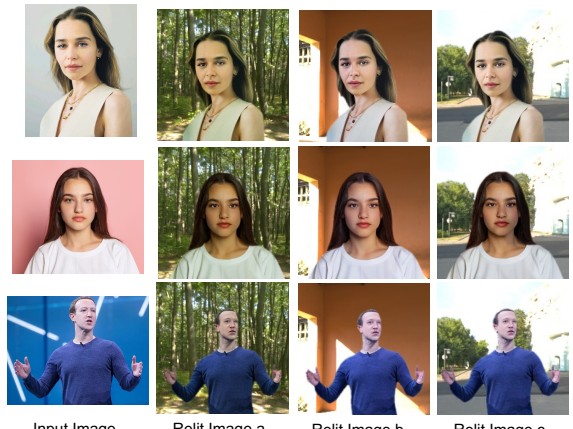

Figure 11: **Human portrait relighting results.** Overall, in most cases, our method shows good generalization to human portraits (see the second row of the figure above). However, there are instances where it produces suboptimal results. For example, in the first row, the model fails to remove the shadow on the neck in the input image. In the failure case shown in the last row, the model does not preserve critical portrait details, such as the shape of the mouth. This issue may stem from the low-resolution limitations discussed in Sec. 5. Further fine-tuning with additional portrait data and at a higher resolution could enhance the model's performance on portrait images.

## C Inherent color ambiguity

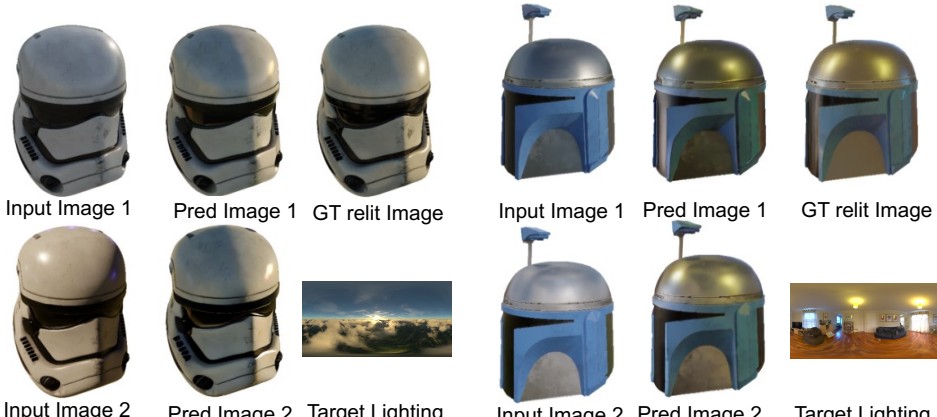

| Input Image 1 | Pred Image 1 | GT relit Image | Input Image 1 | Pred Image 1 | GT relit Image |

| Input Image 2 | Pred Image 2 | Target Lighting | Input Image 2 | Pred Image 2 | Target Lighting |

Figure 12: **Color ambiguity analysis.** We found that our data-driven model can handle the inherent color ambiguity to some extent, especially when the objects' materials in the input image are relatively specular. Because the diffusion model can infer the lighting of the input image through the reflections or specular highlights

Color ambiguity is an inherent issue in the single-image relighting task. That said, we found that our data-driven model can handle this problem to some extent, especially when the objects' materials in the input image are relatively specular. Because the diffusion model can infer the lighting of the input image through the reflections or specular highlights. In Fig. 12, we show two such examples: we render each object with two different environment maps and relight them with the same environment map. In both cases, the input images have different colors and we want to relight them under the same target lighting. It turned out that both input images can be relit to the results that match the ground truth results, which indicates our model has some ability to handle the color ambiguity.

As for the general diffuse object, we think our model could learn an implicit prior over common colors that show up in object albedos, as well as common colors that occur in lighting, and use that to help resolve this ambiguity. However, this may fail sometimes, as we discussed in Sec.5. One potential solution could be also inputting the background of the input image to the diffusion model since the diffusion model can learn to infer the lighting of the input image from the background. We leave this as future work.

# D   More results of relighting 3D objects

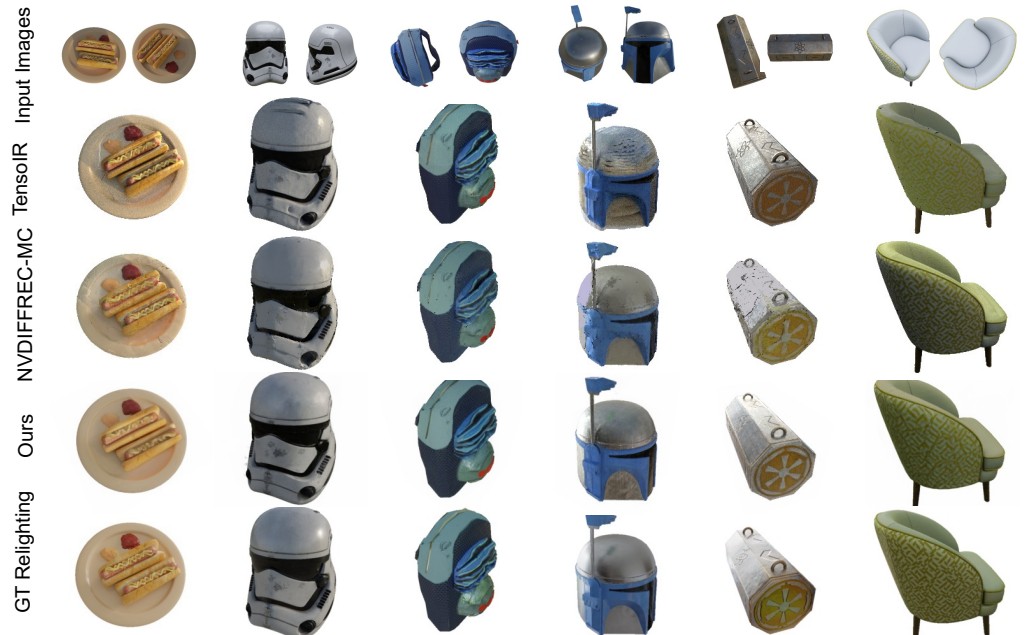

Figure 13: **More results of relighting 3D objects.** We present additional 3D relighting comparison results, which include all of our testing objects. The results show that our methods achieve better qualitative relighting outcomes for all the tested objects

