# OpenReview forum: "Neural Gaffer: Relighting Any Object via Diffusion"
_NeurIPS.cc/2024/Conference — NeurIPS 2024 poster_

### Official Review · Reviewer_C74A · 2024-07-06

**Soundness:** 3
**Presentation:** 3
**Contribution:** 2
**Rating:** 4
**Confidence:** 4

**Summary:**

This paper presents a method for relighting objects observed from a single image. While existing approaches rely on specific capture condition using flashlight illumination or portrait captures, or require to explicitly decompose the scene into geometry and reflectance, the proposed method aims to generate images of a given objects under novel illumination conditions for arbitrary environmental lighting conditions. The authors show that this is possible by relying on a generative diffusion method that is conditioned on the environmental map. The method relies on a pre-trained diffusion model that is fine-tuned on a synthetic relighting dataset to learn the conditioning. The approach is evaluated qualitatively and quantitatively on single-object images. Relying on a conditional diffusion model for relighting, the authors also show additional conditioning on text for relighting.

**Strengths:**

This work presents a simple (this is a good thing) and effective method for relighting from a single image. The method relies on synthetic supervision with a novel Blender-rendered dataset that uses Objeverse as input model source. The authors went a long way by collecting diverse HDR environment maps from the Internet that were augmented to produce a large synthetic relighting dataset of almost 20M rendered images with ground truth lighting maps. Overall, the method offers a number of intriguing benefits listed as follows:

* Conditional image-to-image diffusion model: The method inherits a conditional Zero-1-to-3 model that is extended in its input latents to a rotated environment map with the camera coordinate frame, allowing for image-to-image relighting in a consistent frame. While, given enough training data, the method is effective in relighting, the approach also enjoys the benefits of existing diffusion architectures with various types of conditioning. The authors demonstrate this effectively with their image conditioning.

* Relighting 3D radiance fields: The proposed method is evaluated as a prior for 3D relighting of a neural radiance field. Specifically, the authors propose to use diffusion-based relighting as a coarse reconstruction loss (predicting a coarse relit scene during the NeRF optimization) and a detail refinement loss where the NeRF appearance is further refined.

* Qualitative evaluation: The evaluations presented qualitatively in the main manuscript and the supplemental material in the form of supplemental videos are visually plausible and convincing.

* Quantitative evaluations: The method is adequately ablated and quantitatively compared to single image relighting methods, 3D radiance field relighting with reasonable margins on the test sets. This validates the method as an effective approach.

**Weaknesses:**

What makes the method exciting, at first glance, is also one of the major weaknesses: the technical novelty. The paper piggy-backs on an existing generative method, the Zero-1-to-3 model, that is with a few variations used for relighting. While the simplicity is something that is desired, it also makes it challenging for the reader to derive deeper insights from this work. We learn that pre-trained diffusion-models, when just given enough and the right synthetic data, can allow for plausible novel view synthesis with artefacts that are improved over existing methods. However, the recent work by

Chong Zeng, Yue Dong, Pieter Peers, Youkang Kong, Hongzhi Wu, and Xin Tong. Dilightnet: Fine-grained lighting control for diffusion-based image generation, 2024.

in a way also does show the exactly same, although the technical approach is different. Overall, the technical contribution of the approach is rather incremental (although the method is effective). As such, I am torn on this work. While the technical contribution is not near other work at NeurIPS, the method is effective and likely of high impact.

A further qualm I have is regarding the results compared to NVDIFFREC. While the margins are not substantially different, the results in Fig. 6 seem to indicate differently. It seems as if these results are cherry-picked.

**Questions:**

See questions regarding trends in the quantitative evaluations and the qualitative results that do not seem to match.

**Limitations:**

All major limitations are addressed. The only open limitation not addressed in the manuscript is the runtime. The authors should address and comment on the runtime for their diffusion model.

---

> ### Author Rebuttal · Authors · 2024-08-07
>
> Thank you for your insightful comments and valuable suggestions. We will revise our paper based on your feedback. Here are our responses to your comments:
>
> **1. Technical contribution is incremental considering DiLightNet**
>
> * Although our method and DiLightNet both approach single-image-relighting using diffusion models, our method is fundamentally different from it and provides novel insights to the community. Dilightnet first estimates a coarse mesh from the single image input using an off-the-shelf depth estimator, then uses Blender to render the estimated mesh with several predefined BRDF materials and target lighting conditions, and finally uses the rendered image maps as the relighting conditions to its diffusion model, which means it still involves some explicit geometry, BRDF and lighting modeling in its relighting process, which can be inaccurate or under-expressive. **In contrast, our work is purely end-to-end and shows that relighting for general objects can be totally data-driven without any explicit physically-based decomposition or modeling. To our knowledge, this is a new insight that does not appear in prior work, including Dilightnet**. And as pointed out by Reviewer YxGp, our work “*provides an avenue for relighting without collecting expensive relighting real-world datasets*”.
> * While our method is simple, the technical details, like how HDR environment maps are represented and encoded to be used as the conditions for diffusion models, are non-obvious and important to get right. We have also conducted sufficient ablation studies to validate the effectiveness of our designs.
> * In contrast to DiLightNet, which only explores 2D relighting, we also demonstrate how to apply our relighting diffusion model as data prior to downstream tasks such as 3D relighting. We emphasize that our work is also the first to show that 3D relighting can be achieved without an explicit inverse rendering framework that solves BRDFs. Our 3D relighting pipeline is fundamentally different from any previous inverse rendering-based 3D relighting work and achieves better visual quality. It has the potential to be a new paradigm of 3D relighting.
> * We believe that our simple but effective design is an advantage, making our model robust, more user-friendly (because our model is end-to-end), and easier to scale up (because the training of our model doesn’t rely on any explicit BRDF or geometry information). We are glad to see that all reviewers agree that our methods are effective. In particular, Reviewer rkJr also agrees that our “simplicity is a strength”.
>
> **2. The 3D relighting results may be cherry-picked**
>
> * To demonstrate that the results were not cherry-picked, we present additional 3D relighting comparison results in Figure 8 of the rebuttal PDF, which includes all of our testing objects. The results show that our methods achieve better qualitative relighting outcomes for all the tested objects. We will provide more results in our final revised paper.
> * Regarding the questions about the "trends in the quantitative evaluations and the qualitative results that do not seem to match," we believe this difference arises because per-pixel-based PSNR does not accurately reflect the visual quality of our diffusion-based relighting method. As shown in Table 3 of our main paper, although our PSNR is only slightly higher than that of NVDIFFREC-MC (29.01 versus 28.25), our LPIPS loss is significantly lower (0.040 versus 0.082). Recent research suggests that human perception-based LPIPS loss is a better indicator of visual quality, which explains the superior visual quality we observe in Figure 6 of our paper.
> * To further illustrate previous point, please refer to Figure 9 in the rebuttal PDF. In that figure, our PSNR is only 0.8 dB higher than NVDIFFREC-MC in the first example and just 0.2 dB higher in the second example. Despite these small differences in PSNR, our method demonstrates much better visual quality. Our LPIPS loss is consistently much lower than the baselines in both examples, reinforcing the idea that human perception-based LPIPS loss aligns more closely with visual quality trends.
>
> **3. Runtime for our diffusion model**
>
> Based on our local test, it takes ~0.5 seconds to relight one image with one A6000 GPU.

---

> > ### Author Response · Authors · 2024-08-12
> >
> > Dear Reviewer C74A,
> >
> > We would like to express our deepest gratitude for the time and effort you have dedicated to reviewing our work and for offering such insightful questions. We greatly appreciate your recognition of its effectiveness, potential high impact, convincing visual results, and thorough evaluations.
> >
> > As the discussion period will close on August 13th, we kindly ask whether our responses have sufficiently addressed your concerns. If there are any remaining issues or points that require further clarification, please let us know. We are eager to provide any necessary support and continue the dialogue.
> >
> > Thank you once again for your valuable time and expertise.

---

### Official Review · Reviewer_YxGp · 2024-07-11

**Soundness:** 3
**Presentation:** 3
**Contribution:** 3
**Rating:** 7
**Confidence:** 3

**Summary:**

The paper introduces Neural Gaffer, an end-to-end 2D relighting diffusion model designed for single-image relighting without the need for explicit scene decomposition. Neural Gaffer can synthesize high-quality relit images of any object under novel environmental lighting conditions by conditioning on a target environment map. The model builds on a pre-trained diffusion model, fine-tuning it on a synthetic relighting dataset. The advantages in generalization and accuracy through evaluations on both synthetic and in-the-wild Internet imagery are shown in the paper. Neural Gaffer can be combined with other generative methods for various downstream 2D tasks like objection insertion. The video results presented in the paper are of high quality.

**Strengths:**

1)  Neural Gaffer performs single-image relighting without the need for explicit scene decomposition into intrinsic components like normals and BRDFs. This provides an avenue for relighting without collecting expensive relighting real-world datasets.

2)  The model can generate relit images of various objects under different environmental lighting conditions based on a target environment map. The method takes a single image as an input.

3) The method can be applied to real-world objects with high-quality relighting results and perform various downstream tasks such as object insertion.

**Weaknesses:**

1) In case of the real-world object scenarios, the object may not be always centred and may have complex backgrounds and lighting to start with. The paper does not demonstrate how would the method behave in such cases. How about the objects with high-frequency texture details?

2) Related to 1) there might be multiple objects in a scene. From the results, it seems that the method cannot handle multiple objects from a single image.

3)  The real-world object examples shown in the paper and the video are good but not impressive. It would be more compelling to show faces, humans, animals etc under the lighting conditions to show the generalizability of the method.

**Questions:**

The paper does not demonstrate how the method behaves when the target object is not centred and has complex backgrounds or varied lighting conditions. How does the method perform in such scenarios, especially with objects that have high-frequency texture details?

It appears that the method may struggle with scenes containing multiple objects. Can the authors provide further evaluation or examples to show how the method handles multiple objects in a single image?

While the real-world object examples are good, they are not particularly impressive. Can the authors provide more compelling examples involving faces, humans, or animals under varied lighting conditions to better demonstrate the generalizability of the method? While it's understood that portrait lighting might not be comparable to those methods specifically trained on portraits, it would be good to see the generalizability of the method.

**Limitations:**

The authors have discussed the limitations.

---

> ### Author Rebuttal · Authors · 2024-08-07
>
> Thank you for your insightful comments and valuable suggestions. We will revise our paper based on your feedback. Here are our responses to your comments:
>
> **1. How the method performs when the target is not centered and has a complex background or varied lighting conditions, especially with objects that have high-frequency texture details**
>
> ​	We have implemented an automatic preprocessing script that can first detect the object to be relit using Segment Anything (SAM), segment it out from the background, and finally move the segmented foreground object to the center of the input image based on its bounding box. Indeed, many objects in our real data results are not originally centered (we show centered images in the paper for better visualization, but our preprocessing script doesn’t need centered objects). We will release this preprocessing code along with our full code release.
>
> ​	We have tested our model on real data with complex backgrounds or varied lighting conditions (as shown in the left column of the supplementary webpage, Fig. 1 in the main paper, and Fig. 3, 4 in the rebuttal file). With the aid of our preprocessing method, we can handle real data with complex backgrounds or varied lighting conditions. In the rebuttal PDF, we show additional examples with high-frequency surface texture details (e.g., sheep and dogs in Fig. 3 of the rebuttal file), and observe high-quality relit results with texture details.
>
> **2. Handling multiple objects**
>
> ​	In our main paper, we did assume there is only a single object present when evaluating various methods, which is a common assumption in most previous object-centric 2D relighting work. And most of our training images only contain one object because we train our model on the filtered Objaverse dataset.
>
> ​	However, we find that our method isn’t limited to just single objects. In Fig. 4 of the rebuttal file, we show that our methods still show good generalization ability when tested on real images containing multiple objects.
>
> **3. More real-world testing examples(animals and human portraits).**
>
> ​	As shown in the Fig. 3 of the rebuttal PDF, our methods show good generalization on real-world animal examples, achieving good relit results with high-frequency surface texture details kept.
>
> ​	As shown in the Fig. 6 of the rebuttal PDF, our methods show good generalization on human portraits in most cases (see the examples of the first two columns in the Fig. 6 of the rebuttal PDF). However, as we mentioned in the limitations section of the main paper since our model is trained on object data and we didn’t specially train it with portrait data, sometimes it might not achieve high-quality results when handling human portraits. We show a failure case example in the last columns in Fig. 6 of the rebuttal PDF, where we found that our model failed to keep the facial details of the portrait image, such as the shape of the mouth. We believe that further finetuning our model with more portrait data can help our model achieve better performance on human portrait data.

---

> ### Author Response · Authors · 2024-08-12
>
> Dear Reviewer YxGp,
>
> We would like to express our deepest gratitude for the time and effort you have dedicated to reviewing our work and for offering so many insightful questions. We greatly appreciate your recognition of its effectiveness and high-quality visual results, and praise that *"it provides an avenue for relighting without collecting expensive relighting real-world datasets."*
>
> As the discussion period will close on August 13th, we kindly ask whether our responses have sufficiently addressed your concerns. If there are any remaining issues or points that require further clarification, please let us know. We are eager to provide any necessary support and continue the dialogue.
>
> Thank you once again for your valuable time and expertise.

---

### Official Review · Reviewer_rkJr · 2024-07-13

**Soundness:** 3
**Presentation:** 2
**Contribution:** 3
**Rating:** 5
**Confidence:** 4

**Summary:**

Neural Gaffer presents an approach to object-centric image relighting using diffusion models. The method adapts a pre-trained diffusion model and fine-tunes it on a synthetic dataset designed for relighting tasks. The main feature is its ability to condition the diffusion process on target environment maps, allowing for control over lighting effects.

**Strengths:**

1) Simple yet effective approach: The paper presents a straightforward fine-tuning method for object relighting, similar to zero-1-2-3 shot learning. This simplicity is a strength, demonstrating that complex relighting can be achieved without overly complicated techniques.

2) Powerful data-driven learning: The supervised conditional diffusion model effectively learns to relight objects, highlighting the potential of data-driven approaches in capturing intricate lighting interactions.

3) Competitive results: Based on the presented figures, the method appears to outperform the recent DiLightNet in some aspects. However, this comparison raises some evaluation questions (see questions section for details).

**Weaknesses:**

1) Real-world evaluation: The model is fine-tuned on a synthetic relighting dataset, which might not fully capture the complexity of real-world lighting scenarios. Real-world evaluation is necessary, and there are datasets capturing these effects. The paper is currently missing this evaluation, and there are datasets available for such evaluation [1] OpenIllumination [2] Objects with Lighting or [3] Stanford ORB dataset. These papers have been cited but it is surprising to not see an evaluation of these datasets.

2) Reliance of Environment map: Do you need to supply the environment map for relighting? There is a missing baseline that shows what happens if you condition the target lighting image without a full environment map (only image crops). The Diffusion Light Probe (CVPR 2024) paper indicates that diffusion models are capable of inpainting reliable environment maps and they seem to be implicitly encoded within the model. This baseline will justify why a full environment map is required or necessary for this task.

3) Generalization to scenes: The extent to which the method generalizes to scenes -- not just objects -- is unclear. Evaluating the MIT-multi illumination dataset could shed light on this. The current reliance on explicit environment maps makes it harder to perform on these scenes, but it would be interesting to see if, without explicit environment maps (like suggested above), can you learn to relight and compare on scenes.

4) Evaluation metrics: Recent studies show that PSNR, SSIM, etc. are not consistent with human evaluation. See "Towards a Perceptual Evaluation Framework for Lighting Estimation" (CVPR 2024). These metrics don't tell us much about whether the method is promising as such. A thorough evaluation via user studies or the metrics as defined in the recent paper is currently missing from the paper.

5) Unrealistic results and missing comparisons: The object insertion results look unrealistic, with incorrect shadows that don't match the lighting conditions. Several relevant lighting-aware compositing methods are missing from the comparisons, such as ControlCom [Zhang et al., arXiv 2023], Intrinsic Harmonization [Carega et al., SIGGRAPH 2023], Reshading [Bhattad and Forsyth, 3DV 2022], and ARShadowGAN [CVPR 2020]. The comparison to AnyDoor doesn't make sense as it's not lighting-aware. Including these comparisons would provide a better evaluation of the method's performance against current state-of-the-art techniques.

6) Further, as the papers use off-the-shelf methods to estimate environmental maps (text2light), why not compare with existing 3D object compositing with lighting estimation methods to get a sense of how the proposed methods compare to these tasks -- see Garon et al (CVPR 2019), StyleLight (Wang et al; ECCV 2022) and similar papers? Rendering objaverse objects using lighting estimated from the mentioned or similar methods would help understand the gaps between explicit environment map prediction methods.

7) 3D relighting evaluation: For the 3D relighting setting, according to the Objects with Lighting 3DV 2024 paper, Mitsuba + NeuS is a stronger baseline compared to TensorIR, which is currently missing in the paper.

8) Failure analysis: The paper mentions in the limitations section that the approach might not work for portrait relighting, but it would be interesting to see the kind of failures the diffusion model makes. The current setup lacks experiments in this direction to see what are these failures to encourage future research. Further, the current papers also do not provide any failure examples from Objaverse instances. Is the method perfect on all unseen objects -- detailed analysis is missing as to what objects the proposed methods perform best or worse on. Such analysis helps scope out limitations of the current methods instead of shallow limitations provided in Appendix D.

9) Lack of comparison with simple color matching baselines: The paper doesn't include a comparison with straightforward color adjustment techniques, such as RGB histogram matching between the inserted object and the target scene. This omission raises questions about how much of the method's perceived success in relighting is due to sophisticated light interaction modeling versus simple color transformations. A comparison with such a baseline would help quantify the added value of the diffusion model approach over a simpler method.

**Questions:**

1) Why weren't datasets like OpenIllumination, Objects with Lighting, or Stanford ORB used for evaluation?

2) Have you explored the necessity of full environment maps for relighting?

3) How well does your method generalize to full scenes, beyond individual objects?

4) Given recent findings on the inconsistency of PSNR and SSIM with human perception for lighting tasks, have you considered user study?

5) Why were comparisons with recent lighting-aware object compositing methods (e.g., ControlCom, Intrinsic Harmonization) not included?

6) Have you considered comparing your method with existing 3D object compositing and lighting estimation approaches?

7) Why wasn't Mitsuba + NeuS used as a baseline for 3D relighting, given its reported strength in recent literature?

8) Can you provide a more detailed analysis of failure cases, including examples from Objaverse instances?

9) Can you provide a simple color histogram matching baseline?

10) Comparison with DiLightNet: DiLightNet offers full image generation with background handling, while Neural Gaffer focuses on object relighting. This raises several points:

- Background consistency: How does Neural Gaffer address the background when relighting objects?
- Evaluation scope: Are quantitative evaluations done on the full scene or just the object region? This impacts the interpretation of results.
- Lighting control: DiLightNet allows full-scene control. How comprehensive is Neural Gaffer's approach in comparison?
- User input method: DiLightNet uses radiance hints, Neural Gaffer uses environment maps. How do these compare in terms of user-friendliness and control/precision?
- Shadow and Indirect lighting effects quality: DiLightNet's shadows and indirect effects appear more convincing from their project page. Can you comment on this difference? Can you provide a user study comparing the perceived lighting quality between Neural Gaffer and DiLightNet?

11) How sensitive is your method to the resolution of input environment maps?

**Limitations:**

Somewhat but not fully. See my weakness 8.

---

> ### Author Rebuttal · Authors · 2024-08-07
>
> Thank you for your detailed comments and insight suggestions. We will revise our paper based on your feedback. Here are our responses to your comments:
>
> **1. Evaluating our relighting model on real-world dataset** (in response to weaknesses 1 and question 1)
>
> We evaluate our diffusion model on in-the-wild real data as shown in Fig. 1 of the main paper and the video results in the supplemental material. We note that other reviewers stated that these results are impressive (reviewer **NcDc,** **YxGp)** and convincing (reviewer **C74A**). We have provide more real-world example in our rebuttal file.
>
> We didn’t evaluate our diffusion model on “Objects with Lighting” and “Standford-ORB” because they only provide relit target images under different camera poses from the input image, which can’t be used for evaluating the single-image relighting task.
>
> We didn’t evaluate our diffusion model on “OpenIllumination” because it is a light-stage relighting dataset with different lighting distribution as our methods’s assumption. (Our relighting diffusion model is designed for and trained with general environment maps.) But we are glad to provide our results in our revised paper if the reviewer still wants to see them.
>
> **2. Necessity of full environment maps for relighting** (in response to weaknesses 2 and question 2)
>
> 1. As stated in the abstract, the main focus of our diffusion model is *“taking a single image of any object and synthesizing an accurate, high-quality relit image under any novel* ***environmental lighting condition***. Our task is accurate single-image relighting with a user-defined target environment map, which is a common setting for the single-image relighting task. Relighting the image input without a full environment map is a different task and not the main focus of our work. Therefore, we consider comparison with such a baseline outside the scope of our work.
> 2. That said, our method can be easily combined with other methods to enable relighting without a full environment map. In fact, we have used the paper the reviewer mentions (“The Diffusion Light Probe” (CVPR 2024)) in our object insertion application (see L 230), enabling single-image relighting conditioned on a scene image.
>
> **3. Generalization to Scenes** (in response to weaknesses 3 and question 3)
>
> 1. We want to re-emphasize that the main focus of this paper is object-level single-image relighting (as indicated by the title — “Neural Gaffer: Relighting Any **Object** via Diffusion”) — hence, scene-level relighting is not the research scope of this work. Scene-level and object-level relighting are two distinct tasks with different potential methodology designs, and we leave scene-level single-image relighting to future work.
> 2. Although our model was only trained with synthetic object data, it still shows a degree of generalization ability on scene-level data: As shown in Fig.5 of the rebuttal PDF, we sample some scene images from the MIT multi-illumination dataset (mentioned by the reviewer) and use our diffusion model to relight them with some target environment maps and get some reasonable results: our model can generate shadows in the scenes as shown in the second column of Fig.5 in the rebuttal PDF, and generate reasonable highlights in the scene as shown in the second example of Fig.5 in the rebuttal PDF. We can’t compare with ground truth on this scene dataset because our model was trained with environment maps as conditions, but the MIT-multi illumination dataset doesn’t provide ground truth environment maps. Thank you for your suggestions on how to test our methods without explicit full environment maps. We describe why we need a full environment map for relighting in the previous subsection of this rebuttal reply.
>
> **4.Evaluation metrics and why we don’t conduct user study** (in response to weaknesses 4 and question 4)
>
> 1. The paper the you mentioned studies lighting estimation, which is a different task from relighting. Its conclusion may not be applicable to the relighting task.
> 2. In addition to the PSNR and SSIM metrics you describe, we also compute LPIPS metrics, which are based on human perception and can better reflect visual quality.
> 3. When ground truth results are available, comparing different methods via a user study is not a common way to evaluate accuracy in the relighting community. Most recent relighting papers only evaluate their methods by computing commonly used metrics, such as PSNR, SSIM and LPIPS. For example, all papers (over 10 recent popular works) that have been compared in the recent real dataset benchmarks mentioned by the reviewers (Stanford-ORB, Openillumination, and Objects with Lighting) all evaluate their methods by computing PSNR, SSIM, and LPIPS without a user study. These benchmarks themselves only compare different methods on their dataset by computing quantitative metrics without a user study.
> 4. We appreciate your suggestion regarding the user study. However, due to the limited time for the rebuttal period, we are unable to finish user studies. If you still think a user study is critical for evaluation, please let us know. We will attempt to complete a user study before the end of the discussion period.
>
>
> ----
> Unfinished. Please keep reading the comments.

---

> ### Author Response · Authors · 2024-08-07
>
> **5. Questions related to object insertion** (in response to weaknesses 5，6，9 and questions 5，6，9)
>
> ​	We sincerely appreciate your detailed and insightful suggestions regarding object insertion. Before addressing each of your questions, we would like to clarify that the primary focus of this paper is on relighting, rather than object insertion. In this work, we introduce a robust 2D relighting diffusion model, which can serve as relighting data prior for various downstream tasks and applications in both 2D and 3D. To illustrate its potential, we developed a straightforward object insertion pipeline as one of the 2D application examples. However, we do not consider object insertion to be the core contribution or the main focus of our work.
>
> ​a. Provide more baselines for object insertion comparison, including a simple color histogram matching baseline
>
> Due to limited time and space in the rebuttal PDF, and because object insertion is not the primary focus and core contribution of this work, we have not provided additional comparisons at this time. We did not compare our methods with the histogram-matching baseline because it is too simple and has not been compared in recent work. However, we are willing to include these baselines you mentioned in our revised paper.
>
>
> b. Comparison with existing 3D object compositing approaches
>
> We used DiffusionLight (CVPR 2024) to estimate the environmental lighting of the scene image in our object insertion results, rather than the Text2Light method you mentioned. Since 3D object compositing and lighting estimation are not the tasks of this paper, we did not consider comparing against the methods you mentioned. In theory, our methods could be combined with the lighting estimation techniques you suggested, potentially leading to improved performance in the object insertion task. We appreciate your suggestion and will consider exploring these design choices in future work, but we emphasize that understanding the gaps between different lighting estimation methods is beyond the scope of this paper.
>
> **6. Compare with “NeuS+ Mitsuba” in the 3D relighting task** (in response to weaknesses 7 and question 7)
>
> 1. We have tried to compare with the SOTA methods in our 3D relighting experiments. Specifically, we chose Nvdiffrec-mc and TensoIR for comparison in our paper. According to the Stanford-ORB benchmarks, Nvdiffrec-mc achieves SOTA results among all tested methods. Similarly, TensoIR attains SOTA results based on the OpenIllumination benchmarks.
> 2. We did not compare our methods with Neus+Mitsuba because it is not a commonly used baseline in the field of 3D relighting, and we were not aware of it initially. To our knowledge, Neus+Mitsuba has only been compared in the paper "Objects with Lighting," and we have not seen other recent 3D relighting work that includes this baseline.
> 3. During the rebuttal period, we tested Neus+Mitsuba on our testing dataset using its official code. The results showed that its performance metrics are as follows: PSNR: 26.47, SSIM: 0.912, and LPIPS: 0.073. Therefore, our methods outperform Neus+Mitsuba on all metrics.
>
> **7. Detailed Analysis of Failure Cases** (in response to weaknesses 8 and question 8)
>
> Thanks for your suggestions. We have provided a more detailed analysis of the failure cases below:
>
> a. Portrait Relighting Analysis
>
> Overall, our method demonstrates good generalization to human portraits in most instances (see the first two rows of Fig. 6 in the rebuttal PDF). However, we observed that it sometimes struggles to preserve facial details. For instance, in the failure case shown in the last row of Fig. 6, our model failed to retain crucial details of the portrait, such as the shape of the mouth. We believe that further fine-tuning our model with additional portrait data could improve its performance on such images.
>
> b. Limitations in Objaverse Instances
>
> Color Ambiguity: Our model sometimes fails to resolve inherent color ambiguities, particularly when relighting diffuse objects. As shown in the first row of Fig. 7 in the rebuttal PDF, the model struggles to produce accurate color in the relit results. This issue arises from the inherent color ambiguity, where the color in the input image could be attributed to either the material or the lighting, making it a challenging problem. We believe that including the background of the input image in the diffusion model could help mitigate this issue, as the model could learn to infer the lighting conditions from the background. We consider this a potential avenue for future work.
>
> High-Frequency Reflection Generation: Our model also struggles to generate high-frequency relighting reflections. For example, in the second row of Fig. 7 in the rebuttal PDF, while the model accurately generates low-frequency highlights (as indicated by the red box), it fails to produce high-frequency reflections (as indicated by the green box).
>
> ---
> Unfinished. Please keep reading the following comments.

---

> ### Author Response · Authors · 2024-08-07
>
> **8. Questions related to comparison with DiLightNet** (in response to question 10)
>
> Before we answer your questions, we want to first point out that DiLightNet generates the final image by first generating a relit foreground image with its diffusion model (which is exactly what our relighting model can also do), and then generates the background separately. It generates the background in two ways:
>
> (1)When environment maps are given, DiLightNet will render a background image from the target environment map, and then use the foreground mask to composite the relit foreground image and rendered background image to get the final image.
> We have done the same thing in our paper. As shown in Fig.1 & 4 of the main paper and the supplementary webpage, we also synthesize the background for our relit image and generate a full image with the background.
>
> (2) When environment maps are not given, DlightNet just uses a pre-trained diffusion-based inpainting model(the stable-diffusion-2-inpainting model [Stability AI 2022a]) to inpaint a background for the relit foreground image. Although we didn’t do this in our paper, using a diffusion-based inpainting model to inpaint the background is very easy for our methods to do.
>
> In short, our method can achieve the same full image generation with the background as DiLightNet can.
>
> - **Question:** *Background consistency: How does Neural Gaffer address the background when relighting objects?*
>
>     **Answer:** As mentioned at the beginning of this reply subsection, we render a background image from the target environment map. Please refer to the previous paragraphs for more details.
>
> - **Question:** *Evaluation scope: Are quantitative evaluations done on the full scene or just the object region? This impacts the interpretation of results*.
>
>     **Answer:** When conducting quantitative evaluations, we first use the GT foreground masks obtained during rendering to mask out all background pixels of the testing image and set background pixels to be all 1 (purely white pixels). Then we compute the metrics. This means that we only compare metrics on the relit foreground, which is fair for both methods.
>
> - **Question:** *Lighting control: DiLightNet allows full-scene control. How comprehensive is Neural Gaffer's approach in comparison*?
>
>     **Answer:** We are not sure what full-scene control means here. Could you please give us more explanation for this question?
>
> - **Question:** *User input method: DiLightNet uses radiance hints, Neural Gaffer uses environment maps. How do these compare in terms of user-friendliness and control/precision?*
>
>     **Answer:**
>
>
> 	First, DiLightNet uses radiance hints as its diffusion input, not as the user inputs. Radiance hints are attained by first estimating a coarse mesh of the single image input and using Blender to render the estimated mesh with some predefined and fixed BRDF under the target lighting. To relight an image with DiLightNet, users will first need to specify the lighting information used to render the radiance hints. So the real user inputs of *DiLightNet* are the lighting information, i.e. environment map, etc. Therefore, our method has a similar user input as DiLightNet.
>
> 	Second, our methods are more user-friendly. This is because our method is purely end-to-end — users only need to specify the target environment map, and our relighting diffusion will output the relighting diffusion directly. On the contrary, to relight a single image with DiLightNet, users need to first run a monocular depth estimator to attain the mesh, (which may fail) and then need to use Blender to render the radiance hints and use them as the diffusion inputs. This whole process is more complicated and tricky than our methods.
>
> - **Question:** *Shadow and Indirect lighting effects quality: DiLightNet's shadows and indirect effects appear more convincing from their project page. Can you comment on this difference? Can you provide a user study comparing the perceived lighting quality between Neural Gaffer and DiLightNet?*
>
>     **Answer:**
>
>
> 	DiLightNet’s results may look good separately but they are not accurate when compared with the ground truth relighting results, as we show in the Fig. 4 of our main paper. In addition to visual quality, relighting should also achieve accuracy. As we find in Table 1 of our main paper, our methods achieve more accurate relighting results while maintaining high visual quality.
>
> 	We explain why we didn’t perform a user study in the previous subsection of this rebuttal reply.
>
> ---
>
> Unfinished. Please keep reading the following comments

---

> ### Author Response · Authors · 2024-08-07
>
> **9.How sensitive is the method to environment map resolution** (in response to question 11)
>
> ​	We always resize the target relighting environment map to the resolution required by our diffusion model (256 * 256) before inputting them into our diffusion model. We implemented an energy-preserving resizing function to do this. Therefore, our model is not sensitive to the environment map resolution theoretically. We have also tested conditioning on environment maps with different original resolutions locally (such as 512 $\times$ 1024, 1024 $\times$  2048, and 2048 $\times$  4096), and our local results don’t show obvious differences in diffusion relighting results.

---

> ### Author Response · Authors · 2024-08-12
>
> Dear Reviewer rkjr,
>
> We would like to express our deepest gratitude for the time and effort you have dedicated to reviewing our work and for offering so many detailed and insightful questions. We greatly appreciate your recognition of its effectiveness, competitive results, and powerful data-driven potential.
>
> As the discussion period will close on August 13th, we kindly ask whether our responses have sufficiently addressed your concerns. If there are any remaining issues or points that require further clarification, please let us know. We are eager to provide any necessary support and continue the dialogue.
>
> Thank you once again for your valuable time and expertise.

---

> > ### Comment · Reviewer_rkJr · 2024-08-12
> > **Response to rebuttal**
> >
> > Thanks for the detailed response. Some of my concerns are addressed and I appreciate the author's effort. Some questions remain:
> >
> > 1) One way to evaluate for real-world evaluations might be to use off-the-shelf models to recover environment maps from the relit images -- like DiffusionLight (CVPR 2024) or StyleLight (ECCV 2022). Use these recovered maps as a guide and relight objects and compare them with the GT.
> >
> > 2) Re Necessity of environment maps: My question was about conducting an ablation study to determine if environment maps are really necessary, or if the task can be accomplished using only target image crops. This is because DiffusionLight demonstrates that Stable Diffusion has a strong grasp of environment maps.
> >
> > 3) Re Simple color matching baselines: I don't think there is a satisfactory response to this question. I'm curious to know what would happen if authors simply tried to match the RGB values of the object to the target scene. The easiest way to do this would be to apply a correction coefficient to the object that needs to be relit for each RGB channel. This coefficient would be computed as the average color ratio of the target scene to the source scene. This method provides a good baseline for accounting for global color changes and indicates how much of the result is a result of learning to relight, and how much might be the method finding a shortcut to match the overall color distribution of the target scene.
> >
> > 4) I disagree with the authors who claim that a user study cannot be conducted for this task. In general, for most relighting tasks, there are no reliable test sets with actual ground truth. Also note that these are simulated ground truths, based on approximations from the chosen rendering engines, and should not be assumed to be the actual ground truth.
> >
> > The authors also mention that in the relighting community, it is not common to evaluate accuracy by comparing different methods through a user study when ground truth results are available. However, in object compositing literature, having a user study is still a standard practice even though simulated GT results are available. In addition, the paper already referenced in my review https://arxiv.org/abs/2312.04334 shows that the LPIPS metric is also not reliable contrary to the authors' argument that they adhere to human perception. Please see Fig 5 of their paper. This finding is not only applicable to lighting estimation methods but can also be used to assess how these metrics fare in general for lighting-related tasks. Relying on the argument that previous approaches used certain metrics is not sound when there is now evidence showing that these metrics are unreliable.

---

> > > ### Author Response · Authors · 2024-08-13
> > >
> > > Thanks for your replies. Here are our responses to your remained questions.
> > >
> > > 1. **Suggestions for Real-World Evaluations**
> > >
> > > We appreciate your suggestion to use off-the-shelf models to recover environment maps from the relit images and to use the recovered light maps for relighting and comparison with the ground truth (GT).
> > >
> > > However, we would like to emphasize that relighting results are highly dependent on the input target lighting. To our knowledge, there is currently no off-the-shelf lighting estimation model that guarantees accurate and reliable lighting estimation. Therefore, we believe that using estimated lighting for relighting and then comparing it with GT would not provide a fair and reasonable evaluation for either our method or the baselines. If the relit results do not align with the GT, it would be difficult to determine whether the discrepancy is due to the relighting model or the inaccurate lighting estimation.
> > >
> > > 2. **Re Re Necessity of environment maps**
> > >
> > >     We would like to reiterate that our task is defined as relighting a single image using a full target environment map, which is a common and well-established task setting in many previous single-image relighting papers. While the idea of using incomplete environment maps as a relighting condition is interesting and could be explored in future work, it represents a different task that lies beyond the scope of this paper, and should not be a ablation study.
> > >
> > > 3.  **Re Re Simple color matching baselines**
> > >
> > >     In our rebuttal, we mentioned that we did not include a comparison with simple color histogram matching baselines due to limited time and space in the rebuttal PDF (we had fitted 9 figures on a single page), and the object insertion task in which the simple color histogram matching was suggested to compare was not our primary focus and core contribution. In addition, we did commit to including this comparison in the revised paper.
> > >
> > >     We have now conducted further evaluations using simple color histogram matching baselines to better answer your questions:
> > >
> > >     a. We tested the simple color histogram matching baseline on our 2D relighting dataset. This involved computing the average color ratio between the target relighting background and the input image, then rescaling each RGB channel of the input image accordingly. We then computed metrics such as PSNR, SSIM, and LPIPS, as shown below.
> > >
> > >     b.  We also included the simple color histogram matching baseline in our user studies. (Please refer to the next reply for more details.)
> > >
> > > |  | PSNR↑ | SSIM↑ | LPIPS↓ |
> > > | --- | --- | --- | --- |
> > > | Our Method | 26.706 | 0.927 | 0.041 |
> > > | DiLightNet | 24.823 | 0.918 | 0.056 |
> > > | Simple color matching | 21.894 | 0.912 | 0.072 |
> > >
> > > Both quantitative comparisons and user studies demonstrate that simple color matching performs poorly compared to both our method and the baseline, DiLightNet. This result is intuitive, as relighting involves not only color matching but also complex appearance changes, such as shadow generation, highlight creation, and other nuanced effects. We have illustrated these challenging scenarios in Figures 1 and 4 of the main paper and on our supplementary website.

---

> > > > ### Author Response · Authors · 2024-08-13
> > > >
> > > > **4. Questions of the user study**
> > > >
> > > > a. First, we would like to clarify that no author stated, "*a user study cannot be conducted for this task*." The original statement was, *"However, due to the limited time during the rebuttal period, we are unable to complete a user study. If you still believe a user study is critical for evaluation, please let us know. We will attempt to complete a user study before the end of the discussion period."* And we have now conducted the user study during the discussion period.
> > > >
> > > > b. In addition, we would like to further explain why we didn’t do a user study in the main paper: Our paper primarily focuses on relighting. As we said, most recent relighting papers, including the three relighting benchmarks you mentioned(Stanford-ORB, Objects with Lighting, and OpenIllumination), do not conduct user studies when ground truth (GT) data is available. Object compositing is a distinct task from relighting, so its standard practices do not necessarily apply to the relighting community.
> > > >
> > > > We appreciate your sharing of the recent paper https://arxiv.org/abs/2312.04334. According to its abstract, the paper aims to *"provide a new perceptual framework to help evaluate future **lighting estimation algorithms.** "* The caption of Figure 5, which you mentioned, reads, *"Agreement between the observer scores and the metric scores (columns) for all the **lighting estimation methods** ."* Therefore, we believe the main findings of that paper are directly related to lighting estimation and may not be directly applicable to all lighting-related tasks, such as our relighting task.
> > > >
> > > > c. We have now conducted a user study during the discussion period. We randomly sampled 22 objects from our 2D relighting test dataset and randomly selected a target pose and lighting condition. In our user study form, we presented the original input image, the GT relit image, and the relighting results of three different methods (our method, the baseline DiLightNet, and simple color matching) in random order. We then asked users to select the method most similar to the GT image and the second most similar.
> > > >
> > > > We received 43 responses from different volunteers.
> > > >
> > > > As shown below, most users (over 73.9%) found our results to be the most similar to the GT images and very few users found our results least similar to GT, demonstrating a clear advantage of our method over others according to the user study.
> > > >
> > > > **User study**
> > > >
> > > > |  | Our Method | DiLightNet | Simple color matching |
> > > > | --- | --- | --- | --- |
> > > > | most similar to GT | 73.9% | 21.9% | 4.2% |
> > > > | second most similar to GT | 24.2% | 41.1% | 34.6% |
> > > > | least similar to GT | 1.9% | 37.0% | 61.2% |

---

> > > > > ### Comment · Reviewer_rkJr · 2024-08-13
> > > > >
> > > > > Thank you very much! I'm happy to see a simple baseline and user study. While we still have disagreements about the need for environment maps and real-world evaluations (which were my top two concerns in the original review), I believe these issues can be addressed in future work. Because many of my other concerns and questions were addressed, I am raising my rating to borderline accept. I appreciate all the efforts made by the authors. I hope this discussion will be included in the revision.

---

> > > > > > ### Author Response · Authors · 2024-08-13
> > > > > >
> > > > > > Thank you for your time and feedback. We will revise our paper based on our discussions.

---

### Official Review · Reviewer_NcDc · 2024-07-13

**Soundness:** 3
**Presentation:** 3
**Contribution:** 3
**Rating:** 6
**Confidence:** 4

**Summary:**

The paper proposes a novel method for single-image relighting, which takes an image of an object and a target environmental map as inputs. The authors fine-tune Stable Diffusion on a synthetic relighting dataset to output relit images, conditioning on both the input object image and the target environmental map. The authors show their method outperforms existing baselines. Additionally, the trained relighting model can be applied to downstream tasks such as relighting a neural radiance field and object insertion.

**Strengths:**

- I check the video results in the supplementary video. The visual results are impressive.
- The authors have shown several downstream applications using their trained relighting model, including text-based relighting and object insertion.
- The authors have conducted extensive ablation studies to prove the effectiveness of their proposed method.

**Weaknesses:**

I don’t have many complaints about the paper. I list several potential improvements below:

- In the 3D relighting experiments, it seems unfair to compare with inverse rendering methods such as Nvdiffrec-mc and TensoIR, as they can apply any lighting to the object once the material is recovered, while Neural Gaffer needs optimize for every lighting. On the other hand, I think Neural Gaffer should be combined with these inverse rendering methods and provide priors when recovering material and lighting.
- The extrinsic information is injected by rotating the environmental map. However, it seems intrinsic information is not considered, which means there is an assumed fixed FOV. This could introduce biases in downstream applications and limit the input views in 3D relighting.
- The quantitive comparison with IC-Light is missing.
- The generated image resolution is limited to 256x256.

**Questions:**

The problem of inverse rendering with a single image is inherently ambiguous. For example, the object color in the input image could come from either the material or the lighting. I was wondering about the authors' thoughts on this problem in the context of Neural Gaffer. When relighting an object, there could be multiple possible outputs depending on the decomposition of the material and lighting in the input. Is the probabilistic model of Neural Gaffer able to model this perplexity?

**Limitations:**

Yes

---

> ### Author Rebuttal · Authors · 2024-08-07
>
> Thank you for your insightful comments and valuable suggestions. We will revise our paper based on your feedback. Here are our responses to your comments:
>
> **1. Inherent color ambiguity**
>
> ​	Color ambiguity is an inherent issue in the single-image relighting task.
>
> ​	That said, we found that our data-driven model can handle this problem to some extent, especially when the objects’ materials in the input image are relatively specular. Because the diffusion model can infer the lighting of the input image through the reflections or specular highlights. In Fig.1 of the rebuttal file, we show two such examples: we render each object with two different environment maps and relight them with the same environment map. In both cases, the input images have different colors and we want to relight them under the same target lighting. It turned out that both input images can be relit to the results that match the ground truth results, which indicates our model has some ability to handle the color ambiguity.
>
> ​	 As for the general diffuse object, we think our model could learn an implicit prior over common colors that show up in object albedos, as well as common colors that occur in lighting, and use that to help resolve this ambiguity. However, this may fail sometimes, as shown in the first example of Fig.7 in the rebuttal file. One potential solution could be also inputting the background of the input image to the diffusion model since the diffusion model can learn to infer the lighting of the input image from the background. We leave this as future work.
>
> **2. Neural Gaffer versus other inverse rendering work and using our model for materials recovering**
>
> * Although our method must re-optimize the 3D representation for each new lighting, we would like to emphasize that the re-optimization process is relatively fast: in our implementation, the two-stage 3D relighting process takes just a few minutes on a single GPU.
> * The reviewer suggests that we combine Neural Gaffer with inverse rendering methods. That is a good idea, but our paper intentionally sets out to avoid explicit inverse rendering and decomposition, and explore whether we can successfully build systems using an implicit relighting capability. We first achieve this goal in single-image relighting task, and then we follow the same motivation and build our 3D relighting pipeline. It turns out that such a new pipeline works very well and in theory it has better data-driven and representation potential. Since such a new pipeline is essentially different from previous BRDF reconstruction-based methods and has no similar previous baselines to compare with, we can only compare this pipeline with previous reconstruction-based methods that also work on 3D relighting, such as TensoIR and Nvidiffrec-MC.
> * Thank you for your suggestion about using our model as prior for recovering materials. As we noted in our paper, “*our model can also operate as a strong relighting prior for 3D tasks*” (L 18). Using our model to improve materials recovering is for sure another interesting application. Due to the limited time we have, in Fig. 2 of the rebuttal PDF, we show a simple experiment where we use our model as data prior for recovering materials, demonstrating an improvement in albedo reconstruction quality. In particular, given input images with unknown lighting, we relight each of the input views under three different target environment maps, then use TensoIR to reconstruct BRDFs from the multi-illumination relit images (we use TensoIR because it supports multi-illumination input images). Results show that both NVDIFFREC-MC and the original TensoIR bake shadows present in the original images into the albedo map; in contrast, combining TensoIR with our diffusion prior yields albedo maps with reduced baking of shadows. Note that the current experiment is very simple and has the potential to improve for best quality. We leave this as one of the future work.
>
> **3. Camera Intrinsics are not considered**
>
> * Although the training data was rendered with fixed camera intrinsics, the model shows good generalization ability to cameras with different intrinsics when we test our 2D relighting model using real images from the Internet, which have unknown FOV (as shown in Fig. 1 of the main paper and our supplemental webpage). Training our diffusion model using images rendered with different intrinsics may further improve its performance. We thank the reviewer for the suggestion as an important future direction for scaling up the system.
> * Note that for the 3D relighting application, we first reconstruct a radiance field using the multi-view inputs and then relight the reconstructed radiance field. Since the reconstructed radiance field allows us to do novel view synthesis, we can render novel views with any camera intrinsics, and therefore for the 3D relighting task we can always render novel views with the fixed intrinsics used during training.
>
> **4. Quantitive comparison with IC-Light**
>
> ​	Since IC-light assumes a different lighting input from our method (background image versus environment map), we feel it may not be fair for us to compare with it in the main paper because we have more input lighting information, which is why we only compare with it qualitatively on real data in the appendix and the supplementary videos, in which we have shown an obvious advantage over it. But if the reviewer still thinks quantitive comparison with IC-Light is important, we will provide it in our revised paper.
>
> **5. The image resolution is limited**
>
> ​	We have realized that the relatively low image resolution is one of our methods’s key limitations and mentioned this point in our limitations section. The main reason we don’t try higher resolutions is that we only have limited CPUs to preprocess the data and limited GPUs to train the model.
>
> ​	In theory, our methods can support higher resolutions. We will release all code to encourage future work to improve this point.

---

> ### Author Response · Authors · 2024-08-12
>
> Dear Reviewer NcDc,
>
> We would like to express our deepest gratitude for the time and effort you have dedicated to reviewing our work and for offering so many potential improvement suggestions. We greatly appreciate your recognition of its effectiveness, impressive visual results, and extensive ablation studies.
>
> As the discussion period will close on August 13th, we kindly ask whether our responses have sufficiently addressed your concerns. If there are any remaining issues or points that require further clarification, please let us know. We are eager to provide any necessary support and continue the dialogue.
>
> Thank you once again for your valuable time and expertise.

---

### Author Rebuttal · Authors · 2024-08-07

Dear Reviewers,

Thank you for dedicating your time to review our paper and offering insightful feedback. We sincerely appreciate your efforts to help enhance the quality of our research. We are also pleased to note that all reviewers were supportive of our work:

(a)Recognize our methods are effective and have high-quality results (NcDc, rkJr,  YxGp,  C74A)

(c) Praise our methods outperform the existing methods(NcDc, rkJr, C74A)

(b)Acknowledge our extensive ablation study to prove the effectiveness of the proposed method.(NcDc, C74A)

(c) Praise our methods enable many downstream applications(NcDc, YxGp)

(d)Recognize our methods are simple and effective, which is our strength ( rkJr, C74A)

(e) Acknowledge our methods show good generalization ability on real world data, and have impressive (NcDc, YxGp) and convincing (C74A) results

(f) Acknowledge our methos show potwerful data-driven potential ( rkJr) and provide an avenue for relighting( YxGp)



We thank all the reviewers for your insightful suggestions. Since we have shown a lot of new results as required by reviewers in the rebuttal PDF file. Please zoom in for better visualization when reading it.

---

### Decision · Program_Chairs · 2024-09-25

**Decision:**

Accept (poster)

**Comment:**

The paper proposed a simple yet effective approach for single-image relighting with impressive results on relighting tasks by finetuning image-based diffusion models. The paper further showed applications for relighting and object insertion on real-world examples with adequate evaluations and ablation studies to verify the effectiveness of the proposed approach, demonstrating that end-to-end diffusion-based method can achieve the relighting.  After discussing with SAC, the AC agrees with the opinions of the majority of the reviewers and recommends accepting the paper.

Please add the suggested comparisons (full comparison to NVdiffrec-mc) and evaluations (multiple objects/scenes/human portraits, and better metrics) into the paper.